# The Role of Positron Emission Tomography in Clinical Management of Intraductal Papillary Mucinous Neoplasms of the Pancreas

**DOI:** 10.3390/cancers12040807

**Published:** 2020-03-27

**Authors:** Simone Serafini, Cosimo Sperti, Alessandra Rosalba Brazzale, Diego Cecchin, Pietro Zucchetta, Elisa Sefora Pierobon, Alberto Ponzoni, Michele Valmasoni, Lucia Moletta

**Affiliations:** 1Department of Surgery, Oncology and Gastroenterology, 3rd Surgical Clinic, University of Padua, 35122 Padua, Italy; simone.serafini@ymail.com (S.S.);; 2Department of Statistical Sciences, University of Padua, 35122 Padua, Italy; 3Department of Medicine, Nuclear Medicine Unit, University of Padua, 35122 Padua, Italy; diego.cecchin@unipd.it (D.C.); pietro.zucchetta@unipd.it (P.Z.); 4Department of Radiology, Padua General Hospital, 35121 Padua, Italy

**Keywords:** cystic tumor, International Consensus Guidelines, intraductal papillary mucinous neoplasms, pancreatic neoplasms, positron emission tomography

## Abstract

Intraductal papillary mucinous neoplasms (IPMNs) of the pancreas represent a heterogeneous group of tumors, increasingly diagnosed in clinical practice. An early differential diagnosis between malignant and benign lesions is crucial to patient management and the choice of surgery or observation. The therapeutic approach is currently based on a patient’s clinical, biochemical, and morphological characteristics. The latest published International Consensus Guidelines (ICG) make no mention of the role of metabolic assessments of IPMNs. The aim of this study was to review the current literature, examining the role of 18-fluorodeoxyglucose (FDG) positron emission tomography (PET) in IPMN management. An extensive literature review was conducted according to the 2009 Preferred Reporting Items for Systematic Reviews and Meta-Analyses (PRISMA) guidelines, and 10 articles were analyzed in detail, focusing on the value of PET as opposed to other standard imaging criteria. Data were retrieved on 419 patients. The 18-FDG-PET proved more sensitive, specific, and accurate than the ICG criteria in detecting malignant IPMNs (reaching 80%, 95%, and 87% vs. 67%, 58%, and 63%, respectively). Metabolic assessments may be used as an additional tool for the appropriate management of patients with doubtful imaging findings.

## 1. Introduction

Intraductal papillary mucinous neoplasms (IPMN) of the pancreas are papillary, mucin hyper-secreting lesions that originate from the epithelium of pancreatic ducts [1,2]. They account for 0.5% to 10% of all pancreatic exocrine neoplasms [3,4], but their real incidence is difficult to calculate because most of them are detected incidentally during imaging procedures performed for other medical reasons. Because of their tendency to become malignant, IPMNs are considered as precursors of pancreatic ductal adenocarcinomas. At the time of their diagnosis, they may appear as low-, medium-, high-grade dysplasia adenoma or invasive carcinoma [5]. The differential diagnosis between benign and malignant IPMN is crucial to patient management, particularly for the purpose of scheduling surgery appropriately. In fact, surgery offers an opportunity to prevent and potentially cure pancreatic adenocarcinoma, but it may represent an overtreatment in many cases, with a considerable morbidity and mortality even at experienced centers. Several international guidelines have been published on this topic: they classify IPMNs on the basis of the expected risk of malignancy and indicate which lesions should be resected and which ones should only be kept under observation [6,7,8]. Several clinical, biochemical, and radiological criteria are considered to establish the likelihood of IPMNs being malignant, and any consequent indication for surgery. The features that should prompt a surgical procedure in fit patients are termed “high-risk stigmata” [9]. “Worrisome features” are characteristics that warrant further investigation with endoscopic ultrasonography (EUS) to better stratify the associated risk [10,11]. The latest revision of the International Consensus Guidelines (ICG) [8] defines the presence of obstructive jaundice in patients with cystic lesions of the pancreatic head, enhanced mural nodules ≥5 mm in diameter, and a main pancreatic duct (MPD) size of ≥10 mm as “high risk stigmata”. “Worrisome features” include cysts ≥3 cm in size, enhancing mural nodules <5 mm, thickened and enhanced cyst walls, a MPD 5–9 mm in size, abrupt changes in MPD caliber with distal pancreatic atrophy, lymphadenopathy, high serum levels of carbohydrate antigen (CA)19–9, and a rapid rate of cyst growth (>5 mm in 2 years) [8]. The ICG do not consider metabolic assessment of the lesions among the criteria for the risk stratification of IPMNs. Imaging with positron emission tomography using 18-fluorodeoxyglucose positron emission tomography (18-FDG PET) is based on the increased glucose uptake of neoplastic cells, which overexpress the main cell membrane glucose transporter GLUT-1, resulting in higher uptake of 18-FDG [12]. It is part of the standard diagnostic work-up in many neoplasms, but its role in the management of IPMN is not yet defined. Nonetheless, a growing number of studies focusing on the role of 18-FDG PET in the preoperative assessment of IPMNs have been published over the years. The aim of this review was to examine the recent English literature to clarify the value of 18-FDG PET in IPMN management.

## 2. Materials and Methods 

An extensive review of the English literature was conducted according to the 2009 Preferred Reporting Items for Systematic Reviews and Meta-Analyses (PRISMA) guidelines [13]. A systematic search was performed in PubMed and Scopus on articles published from 1 January, 2000 to 31 October, 2019, using the terms: “pancreatic cyst”, “intraductal papillary mucinous neoplasms”, “pancreatic neoplasms”, and “positron emission tomography”. The related articles function was used to expand the search, and all related abstracts, studies, and citations were analyzed. Three authors screened all titles and abstracts for suitable articles, based on a set of eligibility criteria (Table 1). 

Only human studies written in English were considered. Case reports or small series (<5 patients) were disregarded. The full text of the selected studies was obtained. The manuscripts were examined and the related references were checked for potentially relevant papers not identified by the initial screening process. Articles were rejected if imaging, PET data, or histological diagnoses were not clearly reported, or if it proved impossible to extract the data of interest from the text. The studies included in our review were analyzed in terms of: first author’s name, year of publication, number of patients, histology, and malignant or benign preoperative diagnosis. The cut-off for the maximum standardized uptake value (SUV max) and the different radiological criteria adopted in each study were examined. 

The sensitivity, specificity, positive predictive value (PPV), negative predictive value (NPV), and accuracy of 18-FDG PET, and radiological imaging in differentiating between malignant and benign IPMNs were calculated according to the following formulas: sensitivity = true positive (TP)/[TP + false negative (FN)]; specificity = true negative (TN)/[TN + false positive (FP)]; PPV = TP/[TP + FP]; NPV = TN/[TN + FN] and accuracy = [TP + TN]/[TP + TN + FP + FN]. The total numbers of TP, TN, FP, and FN cases were collected patient by patient from each case series, when reported by the authors. In the event of different SUV max cut-offs or radiological criteria being adopted to discriminate benign from malignant lesions preoperatively, the cut-offs or criteria with the best accuracy were recorded and analyzed. A statistical analysis was performed to compare the summary measures (sensitivity, specificity, accuracy) of the two diagnostic tests (18-FDG PET and ICG criteria). The individual 95% confidence intervals of sensitivity and specificity for each diagnostic test were calculated using the Wilson score method. Pearson’s chi-squared test was used to compare the sensitivities and specificities of 18-FDG PET and of the ICG. *p*-values < 0.05 were considered statistically significant. The 95% confidence intervals for the difference and the ratio between the two sensitivities/specificities were calculated using the Gart and Nam score methods with skewness correction. Statistical analysis was performed using the NCSS 2019 Statistical Software [14].

## 3. Results

Overall, 986 records were identified. After screening the titles and abstracts according to our inclusion and exclusion criteria, 100 full-text articles were assessed for eligibility and 10 studies [15,16,17,18,19,20,21,22,23,24] were ultimately included in our critical appraisal (Figure 1). Patients’ data were collected and analyzed on the basis of their radiological or metabolic diagnostic findings.

### 3.1. Diagnostic Accuracy of 18-FDG PET Scan to Identify Malignant IPMN

All 10 studies examined the role of 18-FDG PET in the preoperative assessment of IPMNs. A preoperative 18-FDG PET was performed for a total of 419 patients with a histological diagnosis of IPMN; 210 of these neoplasms were benign and 209 were malignant. The findings of the single studies are listed in Table 2. The SUV max cut-offs used in the different studies to differentiate benign from malignant lesions ranged from 1.3 to 3.0, with a mean value of 2.3. Several authors [15,16,18,19,24] used a SUV max of 2.5 as the cut-off for malignancy on the strength of previous reports in the literature. Takanami et al. [17] adopted a value of 2.3, based on a receiver operating characteristic curve (ROC) analysis in the cohort study balancing the sensitivity and specificity of different cut-offs. Saito et al. [20] chose a SUV max cut-off of 2.0 based on the number of patients diagnosed with high-grade dysplasia as opposed to invasive carcinoma. Furthermore, these authors acquired a late scan (120 min post-injection) to calculate a Retention Index (RI = (SUV_delayed_ – SUV_early_) × 100/SUV_early_), which further increased the specificity of 18-FDG-PET from 88% to 94%, without any loss of sensitivity. Roch et al. [21] considered as malignant any lesion with a SUV max higher than 3. Yamashita et al. [23] defined as malignant a focal uptake of radiotracer with a SUV max ≥1.3. Ohta et al. [22] did not specify a SUV max cut-off, but considered IPMNs as malignant if they had a greater metabolic uptake than the remnant gland. 

Overall, PET scanning identified 168 TP, 198 TN, 10 FP, 43 FN cases, resulting in a sensitivity, specificity, and accuracy of 79.6% (73.7–84.5), 95.2% (91.4–97.4) and 87.4% (84.2–90.6), respectively. The PPV was 94.4% (90.0–96.9), and the NPV 82.2% (76.8–86.5). The Positive Likelihood Ratio (LR+) was 16.6 (9.0–30.4), and the Negative Likelihood Ratio (LR−) was 0.2 (0.2–0.3).

### 3.2. Diagnostic Accuracy of International Guidelines in Identifying Malignant IPMN

Data on the diagnostic accuracy of the ICG criteria were available for a total of 412 patients with histological diagnosis of IPMN (207 benign and 196 malignant lesions). The findings of the single studies are given in Table 3. The radiological criteria used to differentiate benign from malignant IPMNs in the preoperative work-up varied. Hong et al. [15] identified as malignant a lesion with any of the following features: main duct type, marked dilation of the MPD (≥10 mm), large mural nodules (≥1 cm), large cyst size (≥3 cm), irregular or septate cyst, calcifications, or patulous duodenal papilla. Tomimaru et al. [16] analyzed different radiological aspects (cyst size ≥3 cm, dilation of MPD ≥7 mm, presence/absence of mural nodules); in their series, the presence of mural nodules was the most accurate indicator (86%) of malignancy, regardless of their size. Takanami et al. [17] and Saito et al. [20] respectively identified a dilation of the MPD >5 mm and ≥7 mm as the most accurate radiological criterion of malignancy. Other authors [18,19,21,22,24] classified malignant IPMN according to ICG criteria [6,7], while Yamashita et al. [23] chose MPD ≥10 mm as the cut-off for distinguishing between benign and malignant lesions after performing a ROC analysis. The ICG criteria identified 152 TP, 108 TN, 78 FP, and 74 FN in all, resulting in a sensitivity, specificity, and accuracy of 67.3% (60.9–73.0), 58.1% (50.9–64.9), and 63.1% (58.4–67.8), respectively. The PPV was 66.1% (59.7–71.9), and the NPV 59.3% (52.1–66.2). The LR+ was 1.6 (1.3–1.9), and the LR− was 0.6 (0.5–0.7).

### 3.3. Diagnostic Accuracy Comparison of 18-FDG PET and International Consensus Guidelines in Identifying Malignant IPMN 

The 18-FDG PET shows an overall better performance than ICG criteria in identifying malignant IPMN. The sensitivity of 18-FG PET exceeds by about 18.4% (5.8–33.1) the sensitivity of the ICG criteria. This value increases to 63.9% (45.7–87.6) if the specificity is concerned. The two sensitivities differ by 12.4% (4.1–20.5), and the two specificities by 37.1 (29.5–44.8). Both differences are statistically significant at the 5% level (*p*-value = 0.004 and *p*-value < 0.001). The overall accuracy differs by 24.3% (18.7–29.9). The better performance carries also over to the PPV (94.4% vs. 66.1%) and the NPV (82.2% vs. 59.3%). The positive and negative likelihood ratios of 18-FDG PET (16.6 and 0.2) highlight a much stronger association of 18-FDG PET with the presence/absence of the disease than ICG criteria (1.6 and 0.6). 

### 3.4. Statistical Analyses

Reported in the Appendix A.

## 4. Discussion

IPMNs are intraductal tumors characterized by papillary proliferations of pancreatic mucin-producing epithelial cells [25,26]. They are classified according to their site of origin as: main duct (MD) type; branch duct (BD) type; and mixed type (MT) [27]. IPMNs can be divided histologically into four subtypes: intestinal, pancreatobiliary, oncocytic, and gastric [3,28]. Their prognosis varies greatly, depending on their site of origin and histological characteristics: MD-IPMNs carry the highest risk of malignant degeneration (28–81%), followed by MT-IPMN (20–65%), and BD-IPMN (7–42%) [29,30,31,32,33,34]. Appropriate management of these lesions is still a controversial issue. In 2006, an international conference established the International Consensus Guidelines (ICG), proposing criteria for choosing between surgical and surveillance strategies [6]. These guidelines were subsequently revised in 2012, and again in 2017 [7,8]. Other guidelines were published by the American College of Gastroenterology (ACG) [35], the American Gastroenterological Association (AGA) [36], and the European Study Group on Cystic Tumors of the Pancreas [37] (Table 4). The criteria adopted by these latter guidelines have been examined in a few studies [38,39], but not as extensively as the ICG, which remain the standard reference all over the world. 

The standard radiological work-up proposed in the ICG includes computed tomography (CT), magnetic resonance (MR), and magnetic resonance cholangiopancreatography (MRCP), which are very sensitive in differentiating between benign and malignant lesions, but suffer from a suboptimal specificity [33,40,41]. MRCP is more accurate than CT because it can reveal the anatomy of the pancreatic duct, its connections to the cysts in the branch duct, and any parietal nodules or filling defects [18,42]. Hence, it is considered the gold standard diagnostic procedure. The ICG recommend resection for fit patients with one or more high-risk stigmata. In patients with worrisome features, endoscopic ultrasound (EUS) or contrast-enhanced EUS are recommended for a more accurate detection of any malignant characteristics, such as mural nodules and intracystic structures. EUS also enables the fine needle aspiration of cystic fluid and biopsy for biochemical, genetic, and histological analyses [43,44]. Some drawbacks remain despite improvements in radiological imaging and accuracy of international guidelines. A recent literature review reported that up to 14% of IPMNs without worrisome features or high-risk stigmata were ultimately found to harbor high-grade dysplasia or malignancy on final histology [45]. The question is whether or not surgery is always the best solution for patients meeting the ICG criteria. Uehara et al. [34] described a case series of MD-IPMN patients with a low likelihood of malignancy that prompted them to challenge the recommendation for surgery, whereas three large retrospective studies validated the role of surgery for BD-IPMN [46,47,48]. Not all IPMNs presenting with worrisome features according to the ICG are actually malignant, as well not all small, asymptomatic BD-IPMN are benign [49,50,51]. In other words, we need better predictors, especially in elderly patients, or those unfit for prophylactic surgery. Preoperative metabolic characterization using 18-FDG-PET might improve patients’ risk stratification and management. Moreover, 18-FDG-PET can help in the differential diagnosis of benign and malignant lesions regardless of an IPMN’s morphological subtype. 

In the present literature review, PET achieved an overall sensitivity, specificity, and accuracy of 80%, 94%, and 87%, respectively. Unlike those of the ICG criteria, these values have remained constant over the years, and in the various studies reviewed, whatever the time of publication and standard radiological criteria adopted (Figure 2). The better performance of PET is also supported by a superior PPV (94.4% vs. 66.1%) and NPV (82.2% vs. 59.3%). The positive and negative likelihood ratios of 18-FDG PET (16.6 and 0.2) highlight a much stronger association of 18-FDG PET with the presence/absence of the disease than ICG criteria (1.6 and 0.6).

Pedrazzoli et al. [18], and Baiocchi et al. [19] compared the diagnostic efficacy of 18-FDG PET with that of the Sendai consensus guidelines [7]. Pedrazzoli et al. [18] found that, with a SUV cut-off of 2.5, the sensitivity, specificity and accuracy of PET in detecting histologically-confirmed malignancies were 83.3%, 100%, and 91.36% as opposed to 93.2%, 22.2%, and 61.2% of the ICG criteria. Baiocchi et al. [19] reported similar results, with PET achieving a sensitivity, specificity and accuracy of 83%, 100%, and 96% vs. 100%, 22%, and 43% of the ICG. When Roch et al. [21] compared the diagnostic performance of PET with that of the Fukuoka consensus guidelines [8], the former reached a sensitivity and specificity of 61.5% and 94.6%, while for the latter, the sensitivity and specificity were 92.3% and 27%, respectively. When the ICG criteria were used in combination with PET/CT, this association resulted in a 77.8% sensitivity and 100% specificity for the diagnosis of malignant IPMN [21].

It is important to mention that 18-FDG PET can be liable to false-positive results due to inflammatory disease, as in pancreatitis. Dual time-points acquisition [20] could be useful in this context, because the SUV of malignant cells most often increases over time, whereas inflammatory lesions do not show an increasing uptake in late images. As seen in previous experiences [18,21], PET is an effective procedure for the differential diagnosis of benign and malignant lesions, but less useful than the ICG criteria in predicting the risk of benign IPMNs harboring malignancy. Metabolic assessments of IPMN may be used as an additional tool for the appropriate management of patients with doubtful imaging findings. In this subset of patients, the use of the 18-FDG PET in association with ICG criteria, should improve preoperative diagnostic accuracy for malignant lesions. However, the real usefulness of PET in clinical practice remains controversial because it seems clear that PET/CT improves accuracy in the diagnostic work-up of patients with IPMN, but it also increases the related costs, even if no cost-effectiveness comparisons have been published to date. Moreover, it is difficult to compare the related cost increase with the benefit offered by better patient stratification. Therefore, it seems reasonable to suggest that IPMN management be calibrated according to a patient’s individual surgical risk: the high sensitivity and specificity of PET would be especially important when considering surgery for elderly, comorbid, and/or high-risk patients, when we need to strike the right balance between surgical risk and long-term benefit. It would be less useful for patients fit for surgery in whom resection is warranted according to the ICG criteria. The timing of PET in patients under surveillance remains to be established, but it might reasonably be performed when lesions reveal morphological changes or patients become symptomatic. 

The clinical availability of integrated PET/MR scanners opens new perspectives in this field. The offline fusion of PET and MR images has proven already useful in the diagnosis of pancreatic tumors [52], and a pilot study in pancreatic cancer [53] proved the usefulness of combining various imaging biomarkers (Apparent Diffusion Coefficient, SUVmax, etc.), suggesting a possible role for 18-FDG-PET/MR in the management of IPMN. 

## 5. Conclusions

The limited number of published experiences, the retrospective nature of most studies, the different definitions of SUV cut-offs, and the different guidelines considered all interfere with the interpretation of the results of our review. PET, nonetheless, appears to be a useful diagnostic tool in association with standard clinical and radiological criteria, especially when considering older comorbid patients at high surgical risk. The usefulness of PET in the management of IPMNs should be investigated in larger prospective studies.

## Figures and Tables

**Figure 1 cancers-12-00807-f001:**
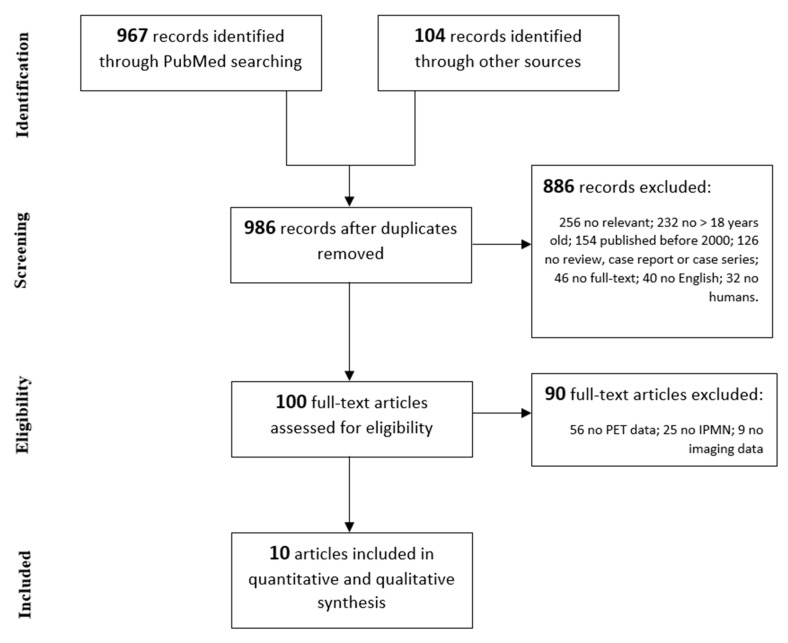
Literature review according to the 2009 Preferred Reporting Items for Systematic Reviews and Meta-Analyses PRISMA guidelines.

**Figure 2 cancers-12-00807-f002:**
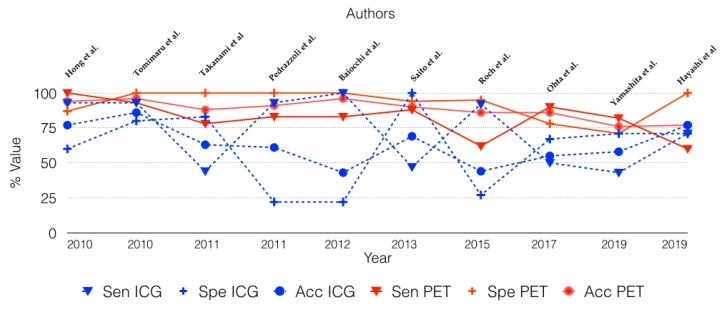
Changes in the sensitivity (SEN), specificity (SPE), and accuracy (Acc) of 18-FDG-PET (PET) and International Consensus Guidelines (ICG) over time.

**Table 1 cancers-12-00807-t001:** Summary of inclusion and exclusion criteria adopted. PET: positron emission tomography; IPMN: intraductal papillary mucinous neoplasms.

Inclusion Criteria	Exclusion Criteria
Articles published from 01/01/2000 to 10/31/2019	Case report or small case series (<5 patients)
Written in English	No PET data available
Study in humans > 18 years old	No radiological imaging data available
	No histopathological proven IPMN

**Table 2 cancers-12-00807-t002:** Summary of 18-FDG-PET results for Intraductal Papillary Mucinous Neoplasm. Tot: number of patients analyzed; Histo: number of patients with histopathological diagnosis available; SUV: Standardized Uptake Value Max; †: SUV max lesion > SUV max normal pancreas.

Author	Year	Patients/Histo	Benign (*n*)	Malignant (*n*)	Sensitivity (%)	Specificity (%)	Accuracy (%)	SUV Cut-Off
Hong et al. [15]	2010	31/31	15	16	100	87	94	2.5
Tomimaru et al. [16]	2010	72/29	15	14	93	100	96	2.5
Takanami et al. [17]	2011	59/16	7	9	78	100	88	2.3
Pedrazzoli et al. [18]	2011	145/69	33	36	83	100	91	2.5
Baiocchi et al. [19]	2012	44/44	32	12	83	100	96	2.5
Saito et al. [20]	2013	48/48	16	32	88	94	90	2.0
Roch et al. [21]	2015	50/50	37	13	62	95	86	3.0
Ohta et al. [22]	2017	29/29	9	20	90	78	86	†
Yamashita et al. [23]	2019	79/38	18	20	82	71	76	1.3
Hayashi et al. [24]	2019	65/65	28	37	60	100	77	2.5
Total		622/419	210	209	80	95	87	

**Table 3 cancers-12-00807-t003:** Summary of diagnostic performance of International Guidelines. Tot: number of patients analyzed; Histo: number of patients with histopathological diagnosis available; †: main duct-type, marked dilatation of the main pancreatic duct (≥ 10 mm), large mural nodule (≥1 cm), large cyst size (≥3 cm), irregular or septate cyst, calcification, or patulous duodenal papilla; MPD: Main Pancreatic Duct; SCG: Sendai Consensus guidelines; FCG: Fukuoka Consensus guidelines.

Author	Year	Tot/Histo	Benign (n°)	Malignant (n°)	Sensitivity %	Specificity (%)	Accuracy (%)	Diagnostic Criteria
Hong et al. [15]	2010	31/31	15	16	93	60	77	†
Tomimaru et al. [16]	2010	72/29	15	14	93	80	86	Mural nodule
Takanami et al. [17]	2011	59/16	7	9	44	83	63	MPD >5 mm
Pedrazzoli et al. [18]	2011	145/80	36	44	93	22	61	SCG
Baiocchi et al. [19]	2012	44/42	30	12	100	22	43	SCG
Saito et al. [20]	2013	32/32	13	19	47	100	69	MPD ≥ 7 mm
Roch et al. [21]	2015	50/50	37	13	92	27	44	FCG
Ohta et al. [22]	2017	29/29	9	20	50	67	55	FCG
Yamashita et al. [23]	2019	79/38	18	20	43	71	58	MPD ≥ 10 mm
Hayashi et al. [24]	2019	65/65	28	37	71	85	77	FCG
Total		622/412	208	204	67	58	63	

**Table 4 cancers-12-00807-t004:** Surgical indication for branch duct (BD), main duct (MD), and mixed type (MT)–IPMN according to most recent guidelines. IPMN: Intraductal papillary mucinous neoplasms; ICG: International Consensus Guidelines; EU: European Study Group on Cystic Tumors of the Pancreas; ACG, American College of Gastroenterology; AGA, American Gastroenterological Association; MPD: main pancreatic duct; CA 19-9: cancer antigen 19-9; DM: diabetes mellitus.

IPMN Type	ICG Guidelines (2017) [8]	EU Guidelines (2018) [37]	ACG Guidelines (2018) [35]	AGA Guidelines (2015) [36]
**BD–IPMN**	**High-risk stigmata:** Enhancing mural nodule ≥5 mm MPD ≥10 mm Jaundice	**Absolute indications:** Solid mass Enhancing mural nodule ≥5 mm MPD ≥10 mm Jaundice	High-risk features: Mural nodule/solid component MPD ≥5 mm MPD caliber change and atrophy Cyst size ≥3 cm Cyst growth ≥3 mm/year Jaundice Acute pancreatitis Elevated serum CA 19-9 New-onset DM	High-risk features: Solid component Dilated MPD Cyst size ≥3 cm
	**Worrisome features:** Growth ≥5 mm/2 years Cyst size ≥3 cm Enhancing mural nodule ≥5 mm Enhancing thickened cyst wall MPD 5–9 mm MPD diameter change Elevated serum CA 19-9 Acute pancreatitis	**Relative indications:** Cyst growth ≥5 mm/year Cyst size ≥4 cm Enhancing mural nodule ≥5 mm MPD 5–9.9 mm Elevated serum CA 19-9 New-onset DM Acute pancreatitis		
**MD/MT-IPMN**	≥1 High-risk stigmata	All	Not mentioned	Not mentioned

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
