# Peer review of "The Role of Positron Emission Tomography in Clinical Management of Intraductal Papillary Mucinous Neoplasms of the Pancreas"

_cancers, 2020, doi:10.3390/cancers12040807_

Round 1

Reviewer 1 Report

Serafini and colleagues presented an interesting review article on the diagnostic usefulness of 18-fluorodeoxyglucose positron emission tomography (18-FDG-PET) for patients with Intraductal Papillary Mucinous Neoplasms (IPMNs) of the pancreas. The authors proposed a kind of meta-analysis analyzing the English articles published from from January 2000 to
October 2019. In particular, they applied inclusion and exclusion criteria thus selecting 10 articles used for further statistical analyses. The results showed that 18-FDG-PET has higher sensitivity, specificity and accuracy in identify IPMNs benign and malign lesions compared to the current ICG diagnostic criteria. Overall, the manuscript is well-conceived, however, there are some critical issues that the authors have to address before publication. Below are reported some minor comments that will improve the quality of the manuscript:
1) In the abstract, please revise the following sentence: “Intraductal papillary mucinous neoplasms (IPMNs) of the pancreas form a heterogeneous group of tumors seen increasingly often in clinical practice.”;
2) The entire manuscript needs English editing performed by an English native speaker;
3) In line 60 use the acronym for 18-fluorodeoxyglucose positron emission tomography;
4) In the Materials and Methods section, please add a Figure reporting the algorithm and inclusion/exclusion criteria used for the selection of papers;
5) In Figure 1, please check the error in the box “886 records excluded”. In particular substitute “pubblished” with “published”;
6) About 90% of the selected studies were excluded from further analyses. What the authors mean with “no center series”? Could these articles be included in the analysis? Please, clarify;
7) Figure S1 is of poor quality. Please provide a supplementary Table or a Figure with a better resolution;
8) The authors should merge the two panels of Figure 2 or add a new figure comparing the sensitivity, specificity and accuracy of 18-FDG-PET with those obtained with ICG criteria (or alternative comparing 18-FDG-PET with computed tomography, magnetic resonance or magnetic resonance cholangio-pancreatography, individually);
9) What is the authors’ opinion regarding the use of both 18-FDG-PET and ICG criteria? In the discussion section, briefly describe the cost-effectiveness of such dual evaluation and the impact of both diagnostic strategies in ameliorating sensitivity, specificity and accuracy;
10) Is this article a resubmission of a previously submitted paper? Please, clarify.

Author Response

Dear Sir,

We would like to thank the reviewer for the constructive comment to our manuscript “The Role of Positron Emission Tomography in the Clinical Management of Intraductal Papillary Mucinous Neoplasms of the Pancreas” (ID: Cancers- 719188).

We herewith provide a point-by-point response to the reviewer’s comments and submit the revised manuscript for possible publication in Cancers Special Issue “Role of Medical Imaging in Cancers”.

  • 1 Reviewer’s comment: In the abstract, please revise the following sentence: “Intraductal papillary mucinous neoplasms (IPMNs) of the pancreas form a heterogeneous group of tumors seen increasingly often in clinical practice.”

Response: as the reviewer suggested, we added in the abstract the sentences “Intraductal papillary mucinous neoplasms (IPMNs) of the pancreas represent a heterogeneous group of tumors, increasingly diagnosed in clinical practice”.

  • 2 Reviewer’s comment: The entire manuscript needs English editing performed by an English native speaker

Response: as the reviewer suggested, the manuscript was checked by a professional English editing service.

  • 3 Reviewer’s comment: In line 60 use the acronym for 18-fluorodeoxyglucose positron emission tomography

Response: as the reviewer suggested, we used the acronym 18-FDG PET.

  • 4 Reviewer’s comment: In the Materials and Methods section, please add a Figure reporting the algorithm and inclusion/exclusion criteria used for the selection of papers

Response: as the reviewer suggested, we added the Table 1 reporting the inclusion and exclusion criteria.

  • 5 Reviewer’s comment: in Figure 1, please check the error in the box “886 records excluded”. In particular substitute “pubblished” with “published”

Response: as the reviewer suggested, we checked the error.

  • 6 Reviewer’s comment: About 90% of the selected studies were excluded from further analyses. What the authors mean with “no center series”? Could these articles be included in the analysis? Please, clarify;

Response: as the reviewer suggested, we changed and clarified the PRISMA flow diagram. In particular we intend “case series” as “center series”. These articles were excluded from analysis according to the small number of patients.

  • 7 Reviewer’s comment: Figure S1 is of poor quality. Please provide a supplementary Table or a Figure with a better resolution

Response: as the reviewer suggested, we provided a text format for better resolution.

  • 8 Reviewer’s comment: The authors should merge the two panels of Figure 2 or add a new figure comparing the sensitivity, specificity and accuracy of 18-FDG-PET with those obtained with ICG criteria (or alternative comparing 18-FDG-PET with computed tomography, magnetic resonance or magnetic resonance cholangio-pancreatography, individually);

Response: as the reviewer suggested, we edited the figure.

  • 9 Reviewer’s comment: What is the authors’ opinion regarding the use of both 18-FDG-PET and ICG criteria? In the discussion section, briefly describe the cost-effectiveness of such dual evaluation and the impact of both diagnostic strategies in ameliorating sensitivity, specificity and accuracy;

Response: as the reviewer suggested, we added the sentences “Metabolic assessments of IPMN may be used as an additional tool for the appropriate management of patients with doubtful imaging findings. In this subset of patients, the use of the 18-FDG PET in association with ICG criteria, should improve preoperative diagnostic accuracy for malignant lesions”. The cost-effectiveness analysis is discussed in lines 372 - 376.

  • 10 Reviewer’s comment: Is this article a resubmission of a previously submitted paper? Please, clarify.

Response: The article is a resubmission of an unreviewed paper. The Editor previously suggested to involve nuclear medicine specialists as co-authors.

We thank the reviewer for the positive comment to our manuscript. We have addressed each of the reviewers’ suggestion and highlighted the revisions using the "Track Changes" function in Microsoft Word.

Kind Regards,

Prof. Cosimo Sperti

Department of Surgery, Oncology and Gastroenterology,

3rd Surgical Clinic, University of Padua,

Via Giustiniani 2, 35128 Padua, Italy

Phone +390498218845

Fax +390498218821

Reviewer 2 Report

The present manuscript describes the potential role of PET imaging, with the 18F-FDG, for the clinical management of patients with intraductal Papillar Mucinous Neoplasms (IPMN) of the Pancreas. 

The authors demonstrated the high sensitivity and sensibility of PET imaging to perform the metabolic assessment of IPMN, and the limit of the study (detection of inflammatory disease of pancreas) is notified. 

The authors open new opportunities for the management of these patients and PET imaging should be considered as a part of the diagnosis, in combination with MRI. 

  • Please increase the quality of the figure S1
  • Be careful using sensibility instead of sensitivity (lines 153-155)  

Due to the potential of PET imaging to diagnose the IPMN, a precursor of pancreatic ductal adenocarcinoma, one of the deadliest cancer, this article should be accepted. 

Author Response

Dear Sir,

We would like to thank the reviewer for the constructive comment to our manuscript “The Role of Positron Emission Tomography in the Clinical Management of Intraductal Papillary Mucinous Neoplasms of the Pancreas” (ID: Cancers- 719188).

We herewith provide a point-by-point response to the reviewer’s comments and submit the revised manuscript for possible publication in Cancers Special Issue “Role of Medical Imaging in Cancers”.

  • 1 Reviewer’s comment: Please increase the quality of the figure S1

Response: as the reviewer suggested, we provided text format for better resolution.

  • 2 Reviewer’s comment: Be careful using sensibility instead of sensitivity (lines 153-155)

Response: as the reviewer suggested, we revised the sentences.

We thank the reviewer for the positive comment to our manuscript. We have addressed each of the reviewers’ suggestion and highlighted the revisions using the "Track Changes" function in Microsoft Word.

Kind Regards,

Prof. Cosimo Sperti

Department of Surgery, Oncology and Gastroenterology,

3rd Surgical Clinic, University of Padua,

Via Giustiniani 2, 35128 Padua, Italy

Phone +390498218845

Fax +390498218821
